# Bereavement care for ethnic minority communities: A systematic review of access to, models of, outcomes from, and satisfaction with, service provision

Catriona R. Mayland[1,2,3]*, Richard A. Powell[4], Gemma C. Clarke[5], Bassey Ebenso[6], Matthew J. Allsop[5]

1 Department of Oncology and Metabolism, University of Sheffield, Sheffield, United Kingdom, 2 Sheffield Teaching Hospitals NHS Foundation Trust, Sheffield, United Kingdom, 3 Palliative Care Department, University of Liverpool, Liverpool, United Kingdom, 4 Department of Primary Care & Public Health, School of Public Health, Faculty of Medicine, Imperial College London, London, United Kingdom, 5 Academic Unit of Palliative Care, Worsley Building, University of Leeds, Leeds, United Kingdom, 6 Nuffield Centre for International Health and Development, Clarendon Way, University of Leeds, Leeds, United Kingdom

* C.R.Mayland@sheffield.ac.uk

**Data Availability Statement:** All relevant data are within the manuscript and its Supporting Information files.

## Abstract

### Objectives

To review and synthesize the existing evidence on bereavement care, within the United Kingdom (UK), for ethnic minority communities in terms of barriers and facilitators to access; models of care; outcomes from, and satisfaction with, service provision.

### Design

A systematic review adopting a framework synthesis approach was conducted. An electronic search of the literature was undertaken in MEDLINE, Embase, PsycINFO, Social Work Abstract and CINAHL via EBSCO, Global Health, Cochrane library, the Trip database and ProQuest between 1995 and 2020. Search terms included bereavement care, ethnic minority populations and the UK setting.

### Results

From 3,185 initial records, following screening for eligibility, and full-text review of 164 articles, seven studies were identified. There was no research literature outlining the role of family, friends and existing networks; and a real absence of evidence about outcomes and levels of satisfaction for those from an ethnic minority background who receive bereavement care. From the limited literature, the overarching theme for barriers to bereavement care was 'unfamiliarity and irregularities'. Four identified subthemes were 'lack of awareness'; 'variability in support'; 'type and format of support'; and 'culturally specific beliefs'. The overarching theme for facilitators for bereavement care was 'accessibility' with the two subthemes being 'readily available information' and 'inclusive approaches'. Three studies reported on examples of different models of care provision.

**Funding:** We did not receive any specific funding to undertake this research study. Dr Catriona R Mayland and Dr Matthew J Allsop are currently funded by the Yorkshire Cancer Research 'CONNECTS' Research Fellowship scheme (award references S406CM and L389MA respectively).

**Competing interests:** The authors have declared that no competing interests exist.

## Conclusions

This review reveals a stark lack of evidence about bereavement care for ethnic minority populations. In particular, understanding more about the role of family, friends and existing support systems, alongside outcomes and satisfaction will begin to develop the evidence base underpinning current provision. Direct user-representation through proactive engagement and co-design approaches may begin to determine the most appropriate models and format of bereavement care for ethnic minority communities to inform service design and delivery.

## Introduction

By 2nd January 2021, 51,437 individuals had died from COVID-19 within the United Kingdom (UK) [1]. A growing concern is the disproportionate impact of the virus on people from ethnic minority communities. This includes impact within both the healthcare sector and in the general population, possibly from situational vulnerabilities (i.e. socio-economic disadvantage, manifesting in factors such as greater exposure to infection and higher prevalence of health vulnerability) [2]. Among over 60,000 excess UK deaths occurring during the pandemic, the highest death rates are recorded among ethnic minority groups [3]. With each decedent potentially affecting at least five others [4], this reflects a substantial number affected by grief and bereavement.

Experiences of death and bereavement are likely to be significantly affected on both an individual and societal level during the pandemic, influenced by factors including the nature of the death, existing family and social support networks and cultural context [5]. Compared with the causes of 'typical' deaths, COVID-19 has distressing symptoms, a rapid progression to the end of life in severe cases, distress related to patient and family isolation (in care settings, and in bereavement) [6], and enforced limitations on meaningful end-of-life rituals. This multiplicity of losses associated with pandemics impacts upon cultural norms, rituals, and usual social practices related to death and mourning, potentially increasing the risk of complicated grief [7].

A recent mapping of ethnic minority mental health services in the UK [8] noted the need to ensure anyone bereaved by COVID-19 receive the appropriate support they need. It was recognised, however, that ethnic minority groups are less likely to access mainstream bereavement services [8]. This may relate to access issues, as well as services not being culturally sensitive or designed in a way to meet the needs of specific communities [9]. One of the key recommendations from this recent report is the need to address gaps in research surrounding ethnicity, bereavement and loss in the UK [8].

In order to identify gaps in research about the existing provision of bereavement support for ethnic minority groups, the public health model [10] provides a conceptual framework aligning interventions with need across three groups. The model proposes that for the majority of individuals, their own inner resources, family and friends, will support their distress and subsequent adjustments to their losses. Approximately a third need non-specialist, structured support. Only a small proportion are at risk of developing 'prolonged grief disorder' (PGD) [11] which involves symptoms of separation anxiety and intense grief, as well as functional impairment which endures for more than six months after the death. The differing components of bereavement support are defined into tiers according to the level of need [12] (Textbox 1):

It is important to acknowledge potential limitations of this public health model from the outset, however, in terms of its applicability to ethnic minority communities. The model was

> ### Textbox 1. Components of bereavement support (National Bereavement Alliance, NBA)
>
> - **Component 1**: most support provided by family, friends, existing networks; information about the experience of bereavement and sign posting to further support is offered on a universal basis (*approximately sixty percent*).
>
> - **Component 2**: individual or group–based structured support sessions (faith groups, befriending groups) for those who are seeking that type of support or are at risk of developing more complex needs (*approximately thirty percent*).
>
> - **Component 3**: specialist interventions provided by specialist counsellors, psychologists or mental health practitioners for those with complex needs, pre–existing mental health conditions or are at high risk of developing prolonged or complicated grief (*approximately ten percent*).

developed by an Australian team of researchers, and initial pilot data involved bereaved carers predominately from an 'Australian' and 'other English speaking' background [13]. Terms such as 'bereavement' and 'grief' are quite specific to the English language and can be difficult to translate into languages which may not have specific words to link with emotions associated with loss or death [14]. Additionally, the model has been developed based on the risk of developing PGD [15], and doesn't fully account for other important sociological or health factors which could impact on need for bereavement support. A pragmatic approach, however, was taken to use this framework to guide initial exploration and mapping of research reporting bereavement support for ethnic minority populations in the UK.

This systematic literature review seeks to synthesize the existing evidence on bereavement care for ethnic minority populations through addressing the following research questions:

1. What are the barriers and facilitators to accessing bereavement care for affected people from ethnic minority populations?

2. What, if any, models of care provision exist to address specific ethnic minority population needs around bereavement?

3. What outcomes are reported for ethnic minority populations accessing bereavement services?

4. What are the levels of satisfaction with bereavement services reported by bereaved people from ethnic minority populations?

In order to clarify what is being described, the following definitions will be used:

Bereavement care: all care provided after the death to those bereaved (all three components of the National Bereavement Alliance (NBA) model) including support from family, friends and existing networks.

Bereavement support: interventions provided at components one and two (provision of information, informal support, individual or group-based structured support).

Bereavement counselling: support provided at component three (specialist interventions).

It is anticipated that findings will clarify the extent of the existing evidence base and potential gaps as well as provide guidance upon which future research, service design and planning, and resource allocation for UK bereavement services can be built.

Ethnic minorities: in the context of this review, we use the UK government guidance on ethnic minorities which refers to all ethnic groups except the White British group. The definition of ethnic minorities is inclusive of White minorities, such as Gypsy, Roma and Irish Traveller groups [16].

It is anticipated that findings will clarify the extent of the existing evidence base and potential gaps as well as provide guidance upon which future research, service design and planning, and resource allocation for UK bereavement services can be built.

## Methods

This review adopted a framework synthesis approach, enabling a structured approach to both organizing and interpreting data.

### Data sources and searches

Databases searched included MEDLINE, Embase, PsycINFO, Social Work Abstracts and CINAHL via EBSCO, Global Health, Cochrane library, the Trip database (inclusive of clinical guidelines, patient information), and ProQuest (e.g. including books, theses and dissertations). This approach reflects a wide range of databases relevant to social science, medical, nursing and allied health professionals and includes grey literature. We conducted database searches on 19th August 2020. The search strategy included terms for bereavement care, ethnic minority populations (including religions commonly practiced by ethnic minority groups in the UK), and terms to focus the review on the UK setting. An example of the search strategy used for MEDLINE is provided (S1 File).

**Study selection.** Studies were included if they reported on bereaved participants of any age (both adults and children) from UK ethnic minority populations or caregivers/family members bereft from death of a person from an ethnic minority group (population). Included studies covered any study design, published in peer-reviewed journals or grey literature presenting primary data. Studies were excluded if they did not report primary data (e.g. opinion pieces, narrative book chapters) specific to ethnic minority populations or were not conducted with a focus on the UK setting. Studies were limited to those published in the last 25 years to reflect contemporary health service delivery structures across the UK.

**Data extraction.** BE, GC and MJA independently screened the titles and abstracts of studies identified. Discrepancies in screening inclusion and exclusion were resolved through discussion. Sample and methodological characteristics to extract were agreed across the team and independently extracted by BE, GC and MJA. Data extracted included: study design, number of participants, summaries of age, gender and ethnicity of study participants, components of bereavement support described, and a summary of key findings.

**Quality assessment.** Due to the heterogeneity of study types, the Mixed Method Appraisal Tool (MMAT) was used for assessing the quality of both quantitative and qualitative studies [17]. All studies were appraised by GC and MJA. Study quality was assessed to appraise included studies but did not inform any exclusion.

**Data synthesis and analysis.** A framework synthesis [18] approach was adopted, incorporating five stages of familiarisation, framework selection, indexing, charting and mapping and interpretation. An initial framework reflecting key components relating to a person's access and interaction with bereavement care across the three levels of the NBA tiers was developed using two sources: literature on known inequities in provision of palliative care (e.g. access, quality and experience of care) [19] alongside expertise across the team in health services and bereavement research. This included four components of: i) access; ii) models and approaches to delivery and provision of bereavement care; iii) outcomes of those accessing bereavement services; and iv) satisfaction with or appraisal of care received. Data from the results and

discussion sections of included articles were independently coded by two authors (MJA, CM) and aligned with the four components of the framework. The framework was adapted and refined during the analysis of data, with documentation of data arising that augmented the definition and content of the four categories. Following alignment of the included studies and meetings between researchers to discuss interpretation and analysis of study findings, an initial conceptual framework was developed. The framework was used to inform engagement with three patient representatives from ethnic minority populations in Bradford and Sheffield and a range of key stakeholders from national advocacy and third sector organisations. This engagement sought to elicit feedback on the framework, guide appraisal of the analysis based on their own experiences and incorporate feedback on alignment of data informed by their expert knowledge. Each key stakeholder also provided specific feedback to guide appropriate terminology. Engagement was conducted by a series of email dialogues and/or video meetings.

## Results

From 3,185 initial records, 3,085 papers were screened for eligibility with full-text reviews of 164 articles reviewed, of which seven were included (Fig 1). These were published across an 18-year period, with two articles published since 2010. Most studies (n = 5) adopted a descriptive observational approach, including a retrospective data analysis [20], questionnaire surveys [21, 22], and face-to-face interviews and focus group discussions [23]. Analytic observational studies were adopted in two studies using comparative cross-sectional questionnaire face-to-face surveys [24, 25]; these two articles report data from the same dataset.

Four of the studies were conducted in London with a focus on specific local authorities [21, 24–26]. Bradford was the location of one study [20]. The remaining two had a national focus with recruitment taking place across England [23], and a study surveying neonatal units across the UK [22].

Studies were focused across secondary and community care settings. Two secondary care studies focused on hospital-based palliative care teams and included one [20] and multiple [21] sites. A third secondary care study focused on hospital-based neonatal teams [22]. Community-based studies included a focus on bereavement among family or close friends of first generation black Caribbean and white populations [24, 25], exploration of bereavement in the context of Gypsy and Traveller communities [23], and accounts of bereaved carers of Bangladeshi patients in east London and their interaction with community-based palliative care [26].

### Quality appraisal

Across all studies, most criteria were met for the relevant MMAT checklists (S2 File). For qualitative studies [23, 26], 100% of quality criteria were reported. For quantitative descriptive studies [20–22], all applicable criteria were met for two studies [20, 22], while it was not possible to determine the relevance of the sampling strategy, sample representativeness or risk of nonresponse bias for a component of a study assessing memorial service invitations [21]. For quantitative non-randomised studies [24, 25], both reporting data from the same dataset, one met all quality criteria [24]. The second of the two quantitative non-randomised studies, which was a short focused report, did not contain information to determine whether confounders were accounted for in the design and analysis [25].

### Participant characteristics

Participants included caregivers, relatives and friends (n = 185) and service providers (n = 389) (Table 1). Where reported, age of caregivers was mostly under 54 years (80% of participants) for two studies reporting data on black Caribbean participants [24, 25] and a median

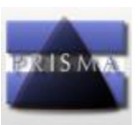

## PRISMA 2009 Flow Diagram

**Identification**

Records identified through
database searching
(n = 3,185)

**Screening**

Records after duplicates removed
(n = 3,085)

Records screened
(n = 3,085)

Records excluded
(n = 2,931)

**Eligibility**

Full-text articles assessed
for eligibility
(n = 154)

Full-text articles excluded,
with reasons
(n = 154)
Not reporting data on
bereavement support in
the UK (n = 106)
Study design not reporting
data (n = 29)
Not focused on ethnic
minorities (n = 12)

**Included**

Studies included in
qualitative synthesis
(n = 7)

*From:* Moher D, Liberati A, Tetzlaff J, Altman DG, The PRISMA Group (2009). *P*referred *R*eporting *I*tems for *S*ystematic Reviews and *M*eta-*A*nalyses: The PRISMA Statement. PLoS Med 6(7): e1000097. doi:10.1371/journal.pmed1000097

**For more information, visit www.prisma-statement.org.**

**Fig 1. PRISMA flow chart outlining screening and inclusion of articles in the review.**

**Table 1. Data on study population extracted from included studies.**

| Citation | Population description | Participant description | Age | Gender | Race/ethnicity composition | How identified as ethnic minorities |
|---|---|---|---|---|---|---|
| [20] | Patients from ethnic minorities referred to the specialist Bradford teaching hospitals palliative care team from 1 October 2001 to 30 September 2002 were identified using the hospital palliative care team database. | 47 patients | Not specified | 30 were male (64%) and 17 were female (36%) | African (n = 1), Afro-Caribbean (n = 5), Bangladeshi (n = 3), Eastern European (n = 8), Indian (n = 6), Pakistani (n = 23) | Review of medical records for demographic information (sex, age, disease, religion and country of origin), the patient's first language, whether they could speak English and whether an interpreter was advised on referral to the team. |
| [24] | Three materially and socially deprived London local authorities—Black Caribbean and White British relatives who registered a death | 100 family members or close friends | Black Caribbean (80% under 54 years) and White British (48% under 54 years) | 72% of Black Caribbean participants were female; 62% of White British participants were female | Black Caribbean (N = 50) White British (N = 50) | Country of origin registered on death certificate—validated at interview |
| [25] | Three materially and socially deprived London local authorities—Black Caribbean and White British relatives who registered a death | 100 family members or close friends | Black Caribbean (80% under 54 years) and White British (48% under 54 years) | 72% of Black Caribbean participants were female; 62% of White British participants were female | Black Caribbean (N = 50) White British (N = 50) | Country of origin registered on death certificate—validated at interview |
| [21] | Staff of bereavement teams based in hospital palliative care teams | 59 nursing staff | Not reported | Not Reported | Not reported | Not applicable |
| [22] | Bereavement teams in neonatal care centres in the UK | 330 staff in neonatal units (87 doctors; 133 nurses; 84 chaplains; 26 not stated) | Not reported | Not Reported | Not Reported | Not applicable |
| [23] | British Romany Gypsies and travellers | 20 bereaved relatives | Only age of relative at death is provided | All participants were female | Irish Travellers (n = 13) and English Gypsies (n = 7) | Via non-governmental organisations (NGOs) that provide advocacy and support to Gypsy/Traveller communities by promoting social inclusion and equality with mainstream society. |
| [26] | Bangladeshi caregivers | 18 bereaved caregivers | Median age of 35, ranging from 25–60 years old | 8 of 18 participants were female | Bangladeshi (n = 18) | Not applicable |

age of 35 for bereaved Bangladeshi caregivers [26]. Ethnic groups represented across the studies included Black Caribbean (N = 50), White British (N = 50), Pakistani (n = 23), Bangladeshi (n = 18), Irish Travellers (n = 13), Eastern European (n = 8), English Gypsies (n = 7), Indian (n = 6), Afro-Caribbean (n = 5), Bangladeshi (n = 3), and African (n = 1).

**Service providers.** Service provider participants included nursing staff based in the bereavement teams of hospital palliative care teams (n = 59), alongside doctors (n = 87), nurses (n = 133), chaplains(n = 84) and not specified (n = 26) participants based in neonatal units across the UK. Age and ethnicity of service provider participants were not reported.

## Key findings from included studies

Table 2 outlines the data on study design, setting and key findings of included studies. The findings, as aligned to four components of the framework used to guide the analysis, are outlined below.

**Table 2.** Data on study design, setting and key findings of included studies.

| Citation | Aim of study | Location | Study design | Setting of study | Study inclusion criteria | Study exclusion criteria | Methods of recruitment | Primary outcome and findings |
|---|---|---|---|---|---|---|---|---|
| [20] | To observe current practice to see if the needs of ethnic minorities were being met by the hospital palliative care team in Bradford, particularly in relation to the addition of the bilingual health-care worker to the team. | Bradford, UK | Retrospective routine data analysis | Acute hospital setting (hospital palliative care team) | Patients from ethnic minorities referred to the specialist Bradford Teaching Hospitals palliative care team | Not specified | Routine data analysis | • Improved care provision: A bilingual health-care worker (BHCW) was involved in 41% of all ethnic minority referrals 53% of South Asian patient referrals to the hospital palliative care team<br>• Continuity of care: As the BHCW worked with both the hospital and community specialist palliative care teams, as well as visiting patients in the hospice, patients could be seen consistently by the same person in different environments.<br>• Bereavement support: the BHCW was involved in bereavement support with one patient's family. By coming from a similar cultural background, the BHCW was able to understand religious beliefs and practice, and how grief is expressed.<br>• Family support: The BHCW was able to provide family support for two other patients. This involved helping to defuse anger with one relative and preventing collusion, when the family did not want the patient to know that the patient had cancer. |
| [24] | To compare the outcomes of bereavement among family or close friends of deceased first-generation black Caribbean and white native-born patients living in the United Kingdom | London, UK | Comparative cross-sectional questionnaire face-to-face survey | Community (three local authorities in London) | 1) Deaths recorded Dec 1997 to Nov 1998; 2) 1st generation black Caribbeans; 3) Progressive or terminal disease (Info from ICD-10 code & 5 causes of death). Comparison group of matched random sample of native-born white patients by age, sex, marital status, diagnosis, and electoral district | Not specified | Selected from death registry and letter sent to relative who registered the death | • Intensity of grief (Core Bereavement Items) was similar between the two groups.<br>• 72 respondents had visited their family doctor subsequent to bereavement, and of these, black Caribbean respondents reported more psychological problems.<br>• Depression and anxiety (GHQ-28) were significantly higher among black Caribbean respondents. Multiple regression analysis revealed this difference was best accounted for by bereavement concerns such as legal and housing problems. |
| [25] | To compare religious faith and support at the end of life of Black Caribbeans and Whites. | London, UK | Comparative cross-sectional questionnaire face-to-face survey | Community (three local authorities in London) | As above, for [22] | Not specified | As above, for [22] | • 96% of deceased Black participants were reported to have either some or a strong religious faith, compared with 58% of White patients.<br>• 87% of Black family members considered that their faith had been a help to them, compared to 51% of White family members. |
| [21] | To describe the provision of bereavement support in the form of a memorial service for relatives of palliative care patients in an acute hospital | London, UK | Questionnaire survey (postal or online not specified) | Acute hospital setting (palliative care services) | Any team identified as a hospital support team providing bereavement support | Excluded any team obviously attached to a hospice, with a bereavement service (BS) or home care (HC). | Questionnaire posted to teams identified through a directory of hospice and palliative care services | • A memorial service replaced bereavement support evenings that had previously been offered by senior members of the oncology and palliative care teams.<br>• Rituals, as part of an integrated service which allow endings, are important care components in the hospital environment. |

(*Continued*)

**Table 2.** (Continued)

| Citation | Aim of study | Location | Study design | Setting of study | Study inclusion criteria | Study exclusion criteria | Methods of recruitment | Primary outcome and findings |
|---|---|---|---|---|---|---|---|---|
| [22] | To establish type of bereavement care services available in neonatal units in the UK and to establish the availability to staff of bereavement education, training, communication, and multicultural support | England, Wales and Northern Ireland | Questionnaire survey (postal or online not specified) | Acute hospital setting (neonatal care units) | Named doctor, named nurse, and a chaplain in each of the 200 neonatal units in England | Not specified | Target respondents were identified through phone contact to neonatal units, requesting the name of attending consultants for a week, the lead nurse or the nurse lead for bereavement care. | • There are deficiencies in staff training and education. Educators must promote the inclusion of content on bereavement/ end-of-life care.<br>• Additional education on cultural issues would be helpful. Managing the bereavement process well to minimize morbidity for families and healthcare providers is an important challenge for the future. |
| [23] | To present a narrative review of the bereavement support needs of Gypsies and Travellers and best practice for organisations seeking to support these populations | England, UK | Face-to-face interviews and focus group discussions | Community setting (Gypsy and Traveller communities) | Women gypsies | Males | Non-governmental organisations (NGOs) that provided advocacy and support to Gypsy/Traveller communities by promoting social inclusion and equality with mainstream society | • Bereavement typically creates long-term problems for Gypsies and Travellers predominant need to 'protect' family at any cost, which has an impact on the complexity of bereavement behaviours of Gypsies and Travellers. It is particularly notable amongst women who will consistently put the care and protection of other family members above their own health and wellbeing.<br>• It is important to recognise that the relatively unchanging family structures and 'traditional' values common to the overwhelming majority of Gypsies and Travellers means that there is an exceptionally high level of contact between kin groups on a daily basis.<br>• Internalisation of feelings and grief makes Gypsies and Travellers particularly vulnerable to poor mental health following significant loss. There is considerable anecdotal evidence that amongst Gypsies and Travellers there are strong cultural taboos associated with acknowledging concerns over mental well-being, with an expectation that people will simply 'get on with it' and men in particular dealing with distress by resorting to drink or 'going off'. Gypsies and Travellers, particularly women, will openly discuss problems they are experiencing with their 'nerves'. However, making other reference to mental illness (including bereavement related depression) is overwhelmingly seen as shameful, with such difficulties typically hidden, dealt with within the family or by self-medicating with drugs and alcohol. |

(*Continued*)

Table 2. (Continued)

| Citation | Aim of study | Location | Study design | Setting of study | Study inclusion criteria | Study exclusion criteria | Methods of recruitment | Primary outcome and findings |
|---|---|---|---|---|---|---|---|---|
| [26] | To increase understanding of the experiences of Bangladeshi caregivers to tailor palliative care services appropriately | London, UK | Ethnographic qualitative study using semi-structured interviews | Community setting (Home of caregiver and hospice) | Carers of all Bangladeshi patients under a community team between 1986 and 1993; Living in the Tower Hamlets area in the east of London | Not reported | Patients were identified by review of all admissions to the team during this period. Case notes were reviewed for demographic data, evidence of communication difficulties, intensity of input of care and any special areas of concern. | • Bangladeshi women had the lowest level of English fluency of all ethnic minority groups which, when combined with their traditional role in this community, resulted in a profound lack of voice and representation as both patients and carers.<br>• No organizational commitment to addressing such issues and the pre-existing model of palliative care was applied to all, regardless of ethnic background.<br>• A lack of bereavement support was revealed. The usual follow-up procedures were inadequate. Travel to the homeland or to be with relatives elsewhere in the UK, incorrect recording of names, communication difficulties and different cultural norms for bereavement all made bereavement follow-up difficult. There were indications of considerable unresolved grief among the carers interviewed.<br>• The large numbers of bereaved children, the majority of whom were under 20 years old, raises important social issues for this community. There were often tensions between generations, precluding adequate support for these children.<br>• Specific bereavement supportive measures are urgently needed. It has been suggested that bereaved children in ethnic minority communities are at particular risk of pathological grief and later maladjustment, given that grief is often compounded by other losses and lack of traditional supports, in a community estranged from its own culture. |

**Access: Barriers and facilitators to bereavement support services.** There were six studies which included some aspect relating to barriers and/or facilitators to accessing bereavement care for people from ethnic minority populations.

**Barriers.** The overarching theme was 'unfamiliarity and irregularities' which related to factors affecting healthcare professionals and/or bereaved ethnic minority communities. There were four identified subthemes including 'lack of awareness'; 'variability in support'; 'type and format of support'; and 'culturally specific beliefs'.

The lack of awareness about bereavement care for ethnic minority communities could arise from factors relating to healthcare professionals. Limited training on bereavement care, perceptions about the training being inadequate (especially around cultural issues) and a dearth of readily available information packs about bereavement care were reported issues [22]. Additionally, medical practitioners' tendency to prescribe medication, rather than provide information and offer psychological counselling, resulted in a lack of awareness about the wider, non-pharmacological support that could be offered for Romany Gypsy and Traveller communities [23].

From the two studies which involved national surveys, services are reported to be variable and often limited. For neonatal units, this was reflected by the variability in both access to interpreting services and the availability of psychological support [22]. For hospital support palliative care teams, over two-thirds (64%) indicated they did not provide any form of bereavement support [21].

The type and format of the bereavement support and counselling services was not always deemed to be needed or suitable for ethnic minority communities. One qualitative study reported that from the 18 participants, 17 found their family was their main source of support during their bereavement [26]. Friends, neighbours and support from their religious communities were also commonplace, with just four indicating that the community specialist palliative care team support had been helpful during this time [26]. A further study, comparing bereaved people from a black Caribbean and white ethnicity backgrounds, emphasised the importance of personal faith and the support provided via their religious leader for the black Caribbean participants [25]. It was also noteworthy that the format of other methods of support, such as a memorial service, traditionally had a strong Christian influence [21]. Whilst this might have been in keeping with the main cultural background from that specific local community, this may have precluded those from other faiths or cultures from attending [21]. Practical legal and financial support was often recognised to be needed rather than specialist interventions. One study indicated that socio-economic factors, such as financial worries, legal and housing problems, were reported more commonly by black Caribbean respondents compared with white respondents, during the bereavement period [24]. These factors were associated with a higher prevalence of anxiety and depression among black Caribbean respondents [24]. A further study also reported on the significant financial difficulties experienced by half of the 19 Bangladeshi participants after the death e.g. trying to meet the costs of transporting the deceased back to Bangladesh [26].

An additional barrier to accessing bereavement care related to culturally specific beliefs. Within Romany Gypsy and Traveller culture, for example, there is a practice of 'not speaking about' bereavement within the close-knit community [23]. This internalisation can potentially increase the risk to poor mental health following significant loss or bereavement. There is reported stigma and shame associated with mental health illness, including bereavement-related depression, resulting in a reluctance to seek formal support [23].

**Facilitators.** The overarching theme was 'accessibility' with the two subthemes being 'readily available information' and 'inclusive approaches'. One study, in the context of support for bereaved parents, reported that information about the needs of different faith groups was

easily available for healthcare professionals to access [22]. Additionally, 78% of participants reported that contact details for representatives of different faith groups were available [22].

A further study, in the context of an invite to a memorial service, suggested that adopting an inclusive invite approach to support was important rather than selecting specific individuals (e.g. only those who had received input from the hospice team) [21].

**Models of care provision.** There were three studies which reported on examples of different models of care provision. One study reported on the role of a bilingual health-care worker in Bradford, based with the local community and hospital-based specialist palliative care teams, to help facilitate links with the South Asian community [20]. Although, the remit was wide, the healthcare worker had a key role in communication and family support prior to death and specifically supported one of the seventeen families during the bereavement period [20]. Their important role in terms of having a similar cultural background, and hence an appreciation and understanding of beliefs, practices and the expression of grief, was also acknowledged [20].

A further study reported on the use of a memorial service for bereaved relatives of palliative care patients who had died within an acute hospital. The aim of this model was to try to address the needs of individuals and families who may not be offered any alternative means of bereavement support [21]. Within the third study, which focused on neonatal units, memorial or remembrance services were also a recognised model of support for bereaved parents [22].

**Outcomes from those who access bereavement services.** There were no identified studies reporting outcomes from ethnic minority groups accessing bereavement support and counselling services. The study focused on the role of the bilingual health-care worker was a retrospective review and no direct views from services users were obtained [20]. It was also noteworthy that no formal evaluation of memorial services had been conducted and that planned evaluations tended to be in the format of audits [21].

**Levels of satisfaction with bereavement services.** There were no identified studies reporting levels of satisfaction with bereavement care, support and counselling ethnic minority groups.

## Conceptual map

An overview of the preliminary conceptual map derived from the study findings (Fig 2) outlines findings from included studies relating to the facilitators, barriers, outcomes and satisfaction relating to bereavement care for ethnic minority groups. This was constructed using the limited research literature available, but it conveys what is available to date.

## Discussion

### Summary of main findings

This review reveals a stark lack of evidence about bereavement care for ethnic minority populations. There is no research literature outlining the role of family, friends and existing networks, other than the suggestion that this type of support, including the role of religious communities and faith, is especially important. From the limited evidence available, there are barriers at each level of the three identified components of bereavement care outlined by the NBA for ethnic minority groups, limiting accessibility. In particular, issues relating to the availability, awareness and dissemination of information were identified, which ideally should be available on a universal basis; furthermore, barriers at components one and two may also impact on awareness and access to bereavement counselling. A lack of relevant, culturally competent training for healthcare professionals can limit access and awareness of potential support services. Additionally, these services may not be structured in a way which meets the

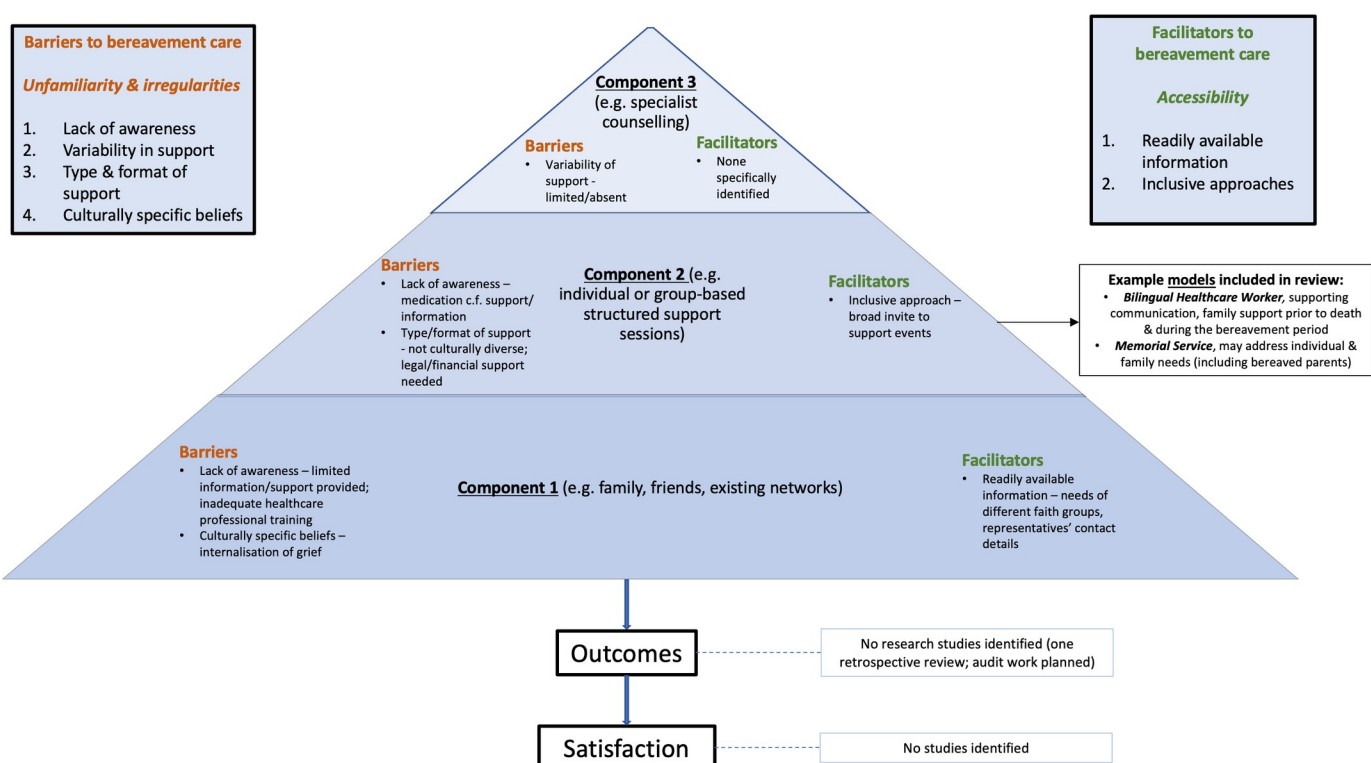

**Fig 2. A preliminary conceptual map of existing facilitators, barriers, outcomes and satisfaction relating to bereavement care for ethnic minority groups based on findings from included studies.**

needs of ethnic minority groups. For example, practical, legal and financial support may be needed and deemed more important by users during the initial bereavement period. A shortage in reporting of facilitators to care was notable, although ensuring readily available information and inclusive approaches (e.g. broad invitations to groups and events) were found to improve accessibility. There were few examples of existing models of care, a real absence of evidence about outcomes and levels of satisfaction for those from an ethnic minority background who receive bereavement care and no identified studies which focused on users who were children.

## Comparison with existing literature

The challenges seen in ensuring adequate access to bereavement care resonates with findings from both the recent ethnic minority mapping report [8] and a previous literature and evidence review focused on racial disparities for mental health services [27]. The latter report found those from black and minority ethnic communities were less likely to access mental health support through their General Practitioner and were more likely to end up in crisis care. Additionally, it is recognised there is a low uptake from ethnic minority groups of Specialist Palliative Care (SPC) services within the UK [28]. Potential reasons are complex and may include a lack of understanding about appropriate referral criteria, lack of awareness or understanding about services, conflicting ideas with the philosophy of hospice and palliative care, previous poor experience, and geographical barriers relating to the location of hospices [28–30].

These findings have wider implications and potential impact on bereavement experience. It is recognised that the quality of care and support prior to death, has subsequent impact on

those who are bereaved [31, 32]. For example, a national post-bereavement survey within Japan, reported that having good end-of-life care discussions was associated with a reduced likelihood of experiencing complicated grief [33]. If access to SPC services is limited for those who are experiencing more complex issues prior to death, these individuals may not receive adequate specialist support in a timely manner both before and after the bereavement. By way of comparison, poor levels of satisfaction with SPC services can be associated with more bereavement related issues [28]. Although SPC services are generally seen as providers of bereavement counselling, they can offer a variety of bereavement care including individual or group-based structured support sessions. Their role, however, should include forming partnerships with relevant agencies and community-based groups to help enable a more comprehensive network for bereavement support. This approach may help 'bridge the gap' in terms of practice and service delivery [34].

Issues relating to the design and structure of services has also been described within a previous study focused on loneliness and isolation for ethnic minority groups [9]. Additionally, social isolation for different ethnic minority older people has also been explored [35]. Despite evidence of need, perceptions from bereaved ethnic minority individuals reflected care was not culturally sensitive or designed with their needs in mind e.g. a preference for informal 'drop in' support [9].

## Implications for research and practice

Future research needs a much greater focus on ethnic minority communities to develop an evidence-base for the provision of bereavement care. In part, this means the recording of relevant data needs to be more comprehensive and complete [36]. Equally, if not more importantly, participatory or co-design methodological approaches are needed [37]. Collaborative studies with key stakeholders from the relevant communities are required to build meaningful alliances and facilitate social change, such as those which have been undertaken with the Roma community to try to overcome health inequalities [38]. Full engagement is needed to ensure culture and beliefs are reflected, respected and guide the development of appropriate bereavement care. Careful consideration as to whom may be key 'door openers' is required. This may include ethnic minority health advocates and faith leaders; building trusting relationships is required.

Key learning from the model of social prescribing [39], which involves link workers providing non-medical support for health and well-being, has relevance in the context of bereavement care. Link workers could provide knowledge and help individuals navigate the different systems and mechanism of support. As well as connections to local community groups, this may also help connections with statutory services providing relevant legal, housing and financial information.

Health and social care agencies aligning with the philosophy and modelling of 'compassionate communities' would be another example of engagement to further enhance bereavement care [34]. Communities are encouraged to 'support people and their families who are dying or living with loss' and provides a linkage between professional healthcare and organic supportive networks within the community [40]. Communication is a key factor in enhancing quality of care. Within the National Ambitions for Palliative and End-of-life Care, the first priority is focused on individualised care and the ability to have honest conversations [41]. It is recognised, however, that those not being able to speak the dominant language e.g. English, can face challenges accessing palliative care. Issues relating to communication between patients, family carers and healthcare professionals can lead to dissatisfaction with care [29]. In the context of a public health approach to support bereavement care, upskilling and enabling relevant

community workers (such as food bank employees, community support workers, hairdressers) [42] to be aware of, and feel competent to provide, component one bereavement support i.e. listening and signposting to additional support may be pertinent. This is not overlooking healthcare professionals, such as General Practitioners, who continue to have a key role to play in minimising barriers to accessing bereavement care. Efforts need to ensure that relevant information is available, accessible, and signposting facilitated to support services which are in keeping with cultural needs.

Further exploration, directly from users, about the most appropriate models and format of support needs to be obtained so that factors important to ethnic minority communities are incorporated into service design and delivery. This links with the recommendations from the recent report [8] focused on the needs of ethnic minority communities arising from the COVID-19 pandemic. These include: i) establishing a national Black, Asian and minority ethnic (BAME) Bereavement Service that meets the cultural needs of the community; ii) bereavement therapists and service providers to have cultural competency training that is quality assured; and iii) the identification of good practice in addressing the disparity of bereavement and loss support for BAME communities using an intersectional approach i.e. understanding further how social, economic and environmental aspects intersect with factors including gender, ethnicity and disability which can promote and disadvantage individuals. In turn, research needs to focus, not just on what is needed, but robust evaluations of the efficacy and satisfaction with different models of care for both adults and bereaved children from ethnic minorities communities. A wider appreciation of the different perceptions, meaning and context of death, dying and bereavement from diverse minority groups is required in order to build an evidence-base of research.

## Strengths and limitations

We adopted a rigorous approach to extracting, searching and appraising the existing literature, working across a multi-disciplinary team and embedding engagement with patient and key stakeholder representatives. Furthermore, we developed a comprehensive search strategy (including terms such as 'refugees') and extracted articles from research databases and those containing grey literature (e.g. Trip database and ProQuest). Included studies comprised descriptive, quantitative non-randomised and qualitative studies. Whilst these study designs are typically aligned with lower levels of evidence [43], we are confident in their findings given a broad appraisal of good quality across the included studies. Our searching of grey literature, however, could have been more extensive if additional hand-searching had been undertaken (potentially using key authors names) and the inclusion of relevant research reported within books. Additionally, we didn't include specific search terms focused on the concept of 'a good death' which may have had relevance about the impact on bereavement care. We focused on UK studies and the literature obtained tended to have a health-based focus. In view of these factors, some sources of data may have been overlooked.

## Conclusions

There is a stark lack of an evidence-base to support UK bereavement care for ethnic minority communities. Our initial, heuristic findings suggest there is a pressing need to understand the role in bereavement played by families, friends, individual religious beliefs and faith networks and other community support systems in aiding those from diverse ethnic minority backgrounds. This needs to be considered in the context of changing demographics and prevalence of specific disease. Fundamental to this agenda will be the effective and meaningful engagement of people from ethnic minority communities in the co-design and interpretation of

future research, to produce findings that can inform and improve service assess, innovate models of service provision and support, and increase satisfaction with those services among the bereaved families.

## Supporting information

**S1 Checklist.**
(DOC)

**S1 File. Example of search strategy used in Medline.**
(DOCX)

**S2 File. MMAT quality appraisal summary.**
(DOCX)

## Acknowledgments

We wish to acknowledge the following individuals for their time and thoughtful contributions as key stakeholders: Jane McCarthy; Alison Penny; Sabeen Zahra; Debjani Chatterjee and Marilyn Relf.

Mr Powell's independent contribution to this article is supported by the National Institute for Health Research Applied Research Collaboration Northwest London. The views expressed in this publication are those of Mr Powell and not necessarily those of the National Institute for Health Research or the Department of Health and Social Care.

## Author Contributions

**Conceptualization:** Catriona R. Mayland, Richard A. Powell, Matthew J. Allsop.

**Data curation:** Catriona R. Mayland, Gemma C. Clarke, Bassey Ebenso, Matthew J. Allsop.

**Formal analysis:** Catriona R. Mayland, Gemma C. Clarke, Bassey Ebenso, Matthew J. Allsop.

**Methodology:** Catriona R. Mayland, Matthew J. Allsop.

**Writing – original draft:** Catriona R. Mayland, Matthew J. Allsop.

**Writing – review & editing:** Richard A. Powell, Gemma C. Clarke, Bassey Ebenso.

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
