## [Decision Letter · Decision Letter 0]

18 Mar 2021

PONE-D-21-04480

Bereavement care for ethnic minority communities: A systematic review of access to, models of, outcomes from, and satisfaction with, service provision

PLOS ONE

Dear Dr. Mayland,

Thank you for submitting your manuscript to PLOS ONE. After careful consideration, we feel that it has merit but does not fully meet PLOS ONE’s publication criteria as it currently stands. Therefore, we invite you to submit a revised version of the manuscript that addresses the points raised during the review process.

We look forward to receiving your revised manuscript.

Kind regards,

Tim Luckett

Academic Editor

PLOS ONE

Journal Requirements:

"There was no specific funding for this study. Dr Catriona Mayland and Matthew J Allsop are

funded by Yorkshire Cancer Research."

**4. **Please include captions for your Supporting Information files at the end of your manuscript, and update any in-text citations to match accordingly. Please see our Supporting Information guidelines for more information: http://journals.plos.org/plosone/s/supporting-information.

Reviewers' comments:

Reviewer's Responses to Questions

**Comments to the Author**

1. Is the manuscript technically sound, and do the data support the conclusions?

Reviewer #1: Yes

Reviewer #2: Yes

2. Has the statistical analysis been performed appropriately and rigorously? 

Reviewer #1: N/A

Reviewer #2: N/A

3. Have the authors made all data underlying the findings in their manuscript fully available?

Reviewer #1: Yes

Reviewer #2: Yes

4. Is the manuscript presented in an intelligible fashion and written in standard English?

Reviewer #1: Yes

Reviewer #2: Yes

5. Review Comments to the Author

Reviewer #1: This article offers an important contribution in terms of identifying the absence of evidence to guide bereavement care and counselling for ethnic minority communities, and provides some justification of the need for further research and investment in participatory development of models of bereavement care. Minor revisions would strengthen and clarify this work, as outlined below.

Introduction

The justification for this review is well positioned at the outset in relation to the COVID-19 landscape. Some suggestions follow:

• Page 5, sentence beginning “Experiences of death and bereavement…” - perhaps insertion of “and are” (or similar) or a comma after “pandemic” would make this sentence clearer to read.

• Page 6: “only a small proportion are at risk of developing ‘complicated grief’…” from O’Connor et al (2014). Some explicit clarification of the meaning of 'complicated grief' as used in the context of this paragraph may be helpful here, ie briefly, how was risk specifically assessed to inform the model cited? More recent conceptualisations of the public health model in the context of bereavement have stratified levels of need according to measures of Prolonged Grief Disorder or PGD, rather than earlier framings and measures of 'complicated grief' eg Aoun et al, 2015 ‘Who Needs Bereavement Support? A Population Based Survey of Bereavement Risk and Support Need’. It may also be worth referring to this in some way, given the current ICD-11 also includes PGD rather than earlier conceptualisations of complicated grief.

• It is valuable and helpful that the article includes acknowledgement of some of the limitations of the public health model. It would be pertinent to also recognise that recent iterations of the model define need according to risk of PGD and thus do not consider complex social/practical issues or other health/mental health concerns in this determination of need – which may be especially relevant given the disproportionate amplification of the risk of these issues among ethnic minority communities.

• Page 7, definitions: I am assuming bereavement is being defined as post-death, however perhaps it would still be helpful to make this explicit given there is often some historical definitional confusion and conflation of terms (e.g. bereavement and grief)

Results

• Inclusion criteria states articles between 2000 and 2020 were included, however the Spruyt paper was published in 1999?

• Page 11: under quality appraisal, the insertion of an extra word may make the second sentence clearer – “…all applicable criteria were met for two studies (19,21), [?while] it was not possible to…”

• In Table 1 and Table 2 the citation numbers appear to be incorrect. In manuscript and reference list, included studies were references 19-25. In tables, citations refer to references 18-24 which seems to be incorrect when looking at reference list.

• Page 19 – briefly, it may be helpful to refer to the nature of the “wider support” suggested for Romany Gypsy and Traveller communities.

• Page 20 – re: the barrier of culturally specific beliefs, perhaps another example (if possible, or was [22] the only relevant article here?) in addition to the example provided that pertained to Romany Gypsy and Traveller communities would thicken the description of this barrier.

• Similarly on page 21, were there any other examples that might enrich the description of the two subthemes for ‘facilitators’?

• Figure 2 – it may be helpful to explicitly articulate the meaning of the arrows on either side of the pyramid on the conceptual map.

Discussion

The point that barriers at lower components may ultimately impact on awareness of and access to bereavement counselling is a valuable one. The call for participatory approaches to research related to ethnic minority communities is also important and well articulated.

• Page 23 – “…practical, legal and financial support may be needed and deemed more important by users during the bereavement period” – literature indicates these forms of support would probably be more significant in ‘early bereavement’ (vs say counselling, which is not often indicated or recommended until later time points). Perhaps it’s useful to clarify this rather than broadly state ‘bereavement period’ – which we know essentially isn’t characterised by a time frame.

• Page 24 – there are other examples of the nature of care experiences impacting upon bereavement and mitigating risk of complex outcomes which may be helpful to reference here to enhance/strengthen this point, eg Garrido, M.M. and Prigerson, H.G. (2014) The End of Life Experience – Modifiable Predictors of Caregivers’ Bereavement Adjustment, Cancer; Anderson, W. G., Kools, S., & Lyndon, A. (2013). Dancing around death: Hospitalist–Patient communication about serious illness, Qualitative Health Research; Ghesquiere, A. Marti Haidar, Y.M., Shear, K. (2011) Risks for Complicated Grief in Family Caregivers, Journal of Social Work in End of Life and Palliative Care (among others).

• Page 24 – Should SPC services be urged to offer bereavement care spanning all three components? What about consideration of the way in which SPC services can partner/collaborate with/support other sectors, agencies and community groups engaged in service provision for ethnic minority communities, to invest in capacity building and support more holistic, comprehensive approaches to bereavement care across the spectrum for non-specialist needs (ie without being the lead provider of direct input across all components)? Would this be an implication that aligns with some of the public health literature and promotes quality of care and support? Further unpacking of some of the issues here may be helpful.

• Page 26 – At the outset, the first and second sentence of second para might benefit from some tightening/clarification – eg who, is being argued, specifically needs to “align” with compassionate communities philosophy?

• Page 26 – sentence starting “It is recognised, however that those not being able to speak…” – after “professionals”, insertion of extra words may assist readability/flow eg “[?and this] can lead to dissatisfaction…”

• Page 27 – the recent COVID-19 report from which the listed recommendations came needs to be referenced here.

• Page 27 – if possible, brief further explanation may be helpful re: what is being suggested by an “intersectional approach,” and why this would be especially relevant here.

Reviewer #2: This review addresses bereavement care for ethnic minority communities in the UK, a woefully under-researched area of bereavement research. Overall, this a well conducted review and will provide a very useful contribution to the literature, highlighting the importance of the needs of ethnic minority communities in the context of the impact of COVID-19 pandemic.

I suggest only a small number of revisions to aid with clarification, described below:

Introduction

- It would be helpful to have in the Introduction a definition of ethnic minorities as the authors are using it in this review, in light of the increasing use of BAME in policy and media. Additionally some details on the context of what communities this might refer to in the UK would be helpful.

- I wonder here whether reference to sociological/anthropological literature on death and bereavement in different cultures might be of use to put into context the gaps in research about the existing provision of bereavement support for

ethnic minority groups e.g.

Death and Bereavement Across Cultures: Second edition edited by Colin Murray Parkes, Pittu Laungani, William Young

Ahluwalia MK, Mohabir RK. Turning to Waheguru: Religious and Cultural Coping Mechanisms of Bereaved Sikhs. OMEGA - Journal of Death and Dying. 2019;78(3):302-313. doi:10.1177/0030222816688907

Dying, Death and Bereavement in a British Hindu Community By Shirley Firth

Yasmin Gunaratnam (2008) From competence to vulnerability: Care, ethics, and elders from racialized minorities, Mortality, 13:1, 24-41, DOI: 10.1080/13576270701782969

Some of this literature may be a bit out of date now but that also reflects problem that understanding cultural practices in minority communities has been limited therefore it is unsurprising bereavement and mental health services continue to fail to meet the needs of communities.

Methods

- p.8-9 Some further details on inclusion and exclusion would be useful for example: was grey literature, commentaries/opinion pieces included? Why was the review limited to publications in past 20 years?

- p.9 Was the full text review and data extraction also completed independently/in duplicate?

- The patient and stakeholder engagement is a really important aspect of the study - could you say a little more about how patients and stakeholders were engaged in the conduct of the review and establishing the framework?

Results

The results are clear and well presented. One suggestion would be to explain the themes and subthemes further to explain what they mean and how they link together. For example the theme ‘unfamiliarity and irregularities’ is this referring to a ‘lack of awareness’ amongst healthcare/bereavement professionals of within bereaved communities or both? Likewise with

‘culturally specific beliefs’.

Figure 2 is a really helpful diagram.

Discussion

- I wonder if the authors could offer any further explanation as to why these barriers exist/persist? Some suggestions are offered on p.24 and it is acknowledged that the reasons are complex. It may that the literature is simply too limited, but in terms of researchers and practitioners being able to take the findings forward, an analysis of some of the potential mechanisms, and not simply an identification of barriers and facilitators would be really valuable.

- It could also be helpful to put the findings in the context of bereavement care as a whole - to what extent are the challenges faced by ethnic minority groups unique and to what extent do they reflect a variable and under resourced sector? Clearly there are unique challenges and it would be helpful to spell these out a little further.

- Finally, could the authors offer any explanation of why the evidence base is so limited? What's going on in UK bereavement research that this is the case? Broader citation of the sociological literature mentioned above might help answer this.

6. PLOS authors have the option to publish the peer review history of their article (what does this mean?). If published, this will include your full peer review and any attached files.

Reviewer #1: No

Reviewer #2: No

---

## [Author Response · Author response to Decision Letter 0]

21 Apr 2021

Have attached file with detailed point-by-point response.

---

## [Decision Letter · Decision Letter 1]

12 May 2021

Bereavement care for ethnic minority communities: A systematic review of access to, models of, outcomes from, and satisfaction with, service provision

PONE-D-21-04480R1

Dear Dr. Mayland,

We’re pleased to inform you that your manuscript has been judged scientifically suitable for publication and will be formally accepted for publication once it meets all outstanding technical requirements, including removal of the repeated paragraph identified by Reviewer 1.

Kind regards,

Tim Luckett

Academic Editor

PLOS ONE

Reviewer's Responses to Questions

**Comments to the Author**

1. If the authors have adequately addressed your comments raised in a previous round of review and you feel that this manuscript is now acceptable for publication, you may indicate that here to bypass the “Comments to the Author” section, enter your conflict of interest statement in the “Confidential to Editor” section, and submit your "Accept" recommendation.

Reviewer #1: All comments have been addressed

Reviewer #2: All comments have been addressed

2. Is the manuscript technically sound, and do the data support the conclusions?

Reviewer #1: Yes

Reviewer #2: Yes

3. Has the statistical analysis been performed appropriately and rigorously? 

Reviewer #1: N/A

Reviewer #2: Yes

4. Have the authors made all data underlying the findings in their manuscript fully available?

Reviewer #1: Yes

Reviewer #2: Yes

5. Is the manuscript presented in an intelligible fashion and written in standard English?

Reviewer #1: Yes

Reviewer #2: Yes

6. Review Comments to the Author

Reviewer #1: The revised submission and responses provided clearly and adequately address the feedback raised. The manuscript illuminates a significant area of concern and represents a valuable contribution to bereavement service development and research.

One minor point – in the clean version of the manuscript, it appears that a paragraph may have been repeated by mistake on page 7: paragraph beginning “It is anticipated that findings will clarify the extent of the existing evidence base…”

Reviewer #2: Thank you to the authors for thoughtful responses to my initial comments. I have no further suggestions and think this is a very valuable review which should pave the way for further improvements serving needs of ethnic minority communities in bereavement care.

7. PLOS authors have the option to publish the peer review history of their article (what does this mean?). If published, this will include your full peer review and any attached files.

Reviewer #1: No

Reviewer #2: No

---

## [Editor Report · Acceptance letter]

8 Jun 2021

PONE-D-21-04480R1 

Bereavement care for ethnic minority communities: A systematic review of access to, models of, outcomes from, and satisfaction with, service provision 

Dear Dr. Mayland:

I'm pleased to inform you that your manuscript has been deemed suitable for publication in PLOS ONE. Congratulations! Your manuscript is now with our production department. 

Kind regards, 

on behalf of

Dr. Tim Luckett 

Academic Editor

PLOS ONE